# Breaking news: Unveiling a new dataset for Portuguese news classification and comparative analysis of approaches

Klaifer Garcia[1], Pedro Shiguihara[2]*, Lilian Berton[1]*

**1** Department of Science and Technology, Federal University of Sao Paulo, São José dos Campos, São Paulo, Brazil, **2** Faculty of Engineering, Universidad San Ignacio de Loyola, Lima, Peru

* pshiguihara@usil.edu.pe (PS); lberton@unifesp.br (LB)

## Abstract

Every day thousands of news are published on the web and filtering tools can be used to extract knowledge on specific topics. The categorization of news into a predefined set of topics is a subject widely studied in the literature, however, most works are restricted to documents in English. In this work, we make two contributions. First, we introduce a Portuguese news dataset collected from WikiNews an open-source media that provide news from different sources. Since there is a lack of datasets for Portuguese, and an existing one is from a single news channel, we aim to introduce a dataset from different news channels. The availability of comprehensive datasets plays a key role in advancing research. Second, we compare different architectures for Portuguese news classification, exploring different text representations (BoW, TF-IDF, Embedding) and classification techniques (SVM, CNN, DJINN, BERT) for documents in Portuguese, covering classical methods and current technologies. We show the trade-off between accuracy and training time for this application. We aim to show the capabilities of available algorithms and the challenges faced in the area.

## Introduction

Internet news is a valuable source of information about a wide range of events. Few events go unnoticed due to the large volume of news published, especially if we consider the variety of news channels, which can be national, regional, or even specialized in a specific topic such as finance or sports. In Google News, the news aggregation tool alone monitors more than 50,000 news sources [1]. The use of this volume of information in automatic text classification has been widely studied, with examples of sentiment analysis applications [2, 3], fake news identification [4, 5], word sense disambiguation [6] and categorization [7].

Recently, we have observed an advance in language modeling techniques through the use of deep learning networks such as Transformers [8] and Convolutional Neural Networks [9] (CNN). These networks require an extensive amount of training data because they have lots of calibration parameters. An example is the BERT network [10], which corresponds to the Encoder of a Transformer, and which was trained with texts extracted from books and Wikipedia. Both are large unlabeled datasets, that are simpler to produce than labeled data but have

**Data Availability Statement:** The dataset is available in this link (https://github.com/Klaifer/PortugueseNewsDataset) The code is available in

this link (https://github.com/Klaifer/
PortugueseNewsCategorization).

**Funding:** This work was financially supported by
Universidad San Ignacio de Loyola in the form of a
grant awarded to PS, and by Conselho Nacional de
Desenvolvimento Científico e Tecnológico (CNPq)
in the form of a grant awarded to LB.

**Competing interests:** The authors have declared
that no competing interests exist.

a limited application. For this reason, a common approach is to divide training into two stages. In the first stage, unlabeled data are used, with a self-supervised approach, so that the network absorbs language knowledge. Self-supervised training is done using the data in a problem that allows the automatic generation of pairs of input and output data. One example, which was used in the BERT network, is the inference of hidden words from an input sentence. The second stage, also called fine-tuning, uses labeled data to specialize the network in an application.

Thus many applications still require labeled data for training. News categorization is one such application and consists of techniques for evaluating the content of articles, grouping them by subject into a known set of categories. This type of application can be utilized in a variety of contexts, such as news aggregation tools that organize news on related subjects like politics and sports. It can also be used to gather data, to be analyzed by other natural language processing techniques, to understand events such as natural disasters [11], epidemic monitoring [3, 12–14], or business events [15–17].

There are some datasets for news categorization in English, however, the number of works produced in Portuguese is significantly lower, and we believe that the categorization of news in Portuguese lacks a comparison of approaches that incorporates these new technologies. For this reason, in this work, we present a comprehensive comparison, starting with traditional feature extraction and classification techniques and progressing to the most recent techniques. Exhaustive evaluations are important to be performed on different data, to understand the capacity of actual text mining techniques.

The main contributions of this work are twofold: i) a comparison of different text representation (BoW, TF-IDF, Embedding) and classification (SVM, DJINN, CNN, BERT) methods for Portuguese news, exploring parameterization and processing time; ii) The creation of a new dataset to evaluate the categorization of Portuguese news. Since there is a lack of Portuguese datasets we aim to encourage other scientists to test their text-processing techniques in this language.

The paper is organized as follows. In Section Related work we list some studies that attempt to categorize texts, particularly those that address the classification of news or texts in Portuguese. We are proposing a new Portuguese news dataset and in Section News dataset are described the procedures we apply for its construction. The methods we used to compare the classifiers are described in Section Experimental setup, including information about the datasets, feature extraction, and the classifiers considered. The comparison of results is presented in Section Results and discussion and we end in Section Conclusions.

## Related work

News categorization is an ongoing research area. In [7] authors compared different classifiers for news categorization, and in [18] authors presented a news classifier using BERT. If we expand our search to include studies in languages other than English, we can find works like [19], which used convolutional neural networks to categorize Chinese news. [20] also classified Chinese news with traditional machine learning algorithms. A comparison of multiple strategies for feature extraction and classification of Roman-Urdu news is presented in [21]. Machine learning techniques, such as Naive Bayes, SVM, and Neural Networks are employed for Nepali news classification [22] and an evaluation of different Deep Learning models for Arabic text classification is presented in [23].

For the Portuguese language, [24] compared BERT to a traditional classifier for news categorization. The experiments, however, are limited to a single classifier configuration and a single dataset, with insufficient data for replication. Including other text classification works, we can mention [25] which presented a method to identify offensive comments on the news. In

this work, the lack of labeled data in Portuguese is mentioned and a technique for building databases on offensive comments in Portuguese is presented. The lack of databases is also mentioned in [26], which presented a technique for identifying text entailment in Portuguese.

Although the absence of labeled data remains a difficulty in many applications, at least for the extraction of semantic features we have observed progress with the emergence of self-supervision learning techniques. These techniques are able to learn semantic features from large sets of unlabeled texts, which are abundant in many languages, exploring problems such as masked word prediction. The word embeddings, like word2vec [27], transformers [8] and neural networks, like BERT, are examples of these self-supervised algorithms.

Considering the self-learning techniques, [28] compared the performance of several word-embedding methods trained for Portuguese, evaluating semantic and syntactic similarities and also the performance in machine learning tasks (part of speech tagging and sentence semantic similarity). In [29] authors presented an approach to named entity recognition using BERT. In this study, a pre-training was applied with data from the brWaC corpus, which is composed of 2.68 billion tokens from 3.53 million documents. In [30], the authors compared the performance of multi-language modeling techniques with techniques specialized in Portuguese and found evidence that specialized models can achieve better performance by better-absorbing knowledge and cultural characteristics of the language.

## News dataset

The lack of labeled training data is a common challenge in natural language processing works, especially when texts in languages other than English are required. In our research, Folha [31] was the most relevant dataset identified for news categorization, which is composed of 167,053 articles unevenly distributed among 48 categories, all extracted from the *Folha de São Paulo* news channel, which has great repercussion in Brazil.

Despite the large number of news in this dataset, they were all collected from a single channel, and other sources would be useful to avoid any bias in news selection or writing style. For this reason, we propose creating a new dataset containing documents collected from Wiki-News, an open-access news channel similar to Wikipedia where content can be collaboratively updated. In this portal, the news is organized into categories such as science, culture, and sports and we can use these as labels for classification problems.

We extracted the data using the Wikimedia dump service [32] and performed the procedures described in Fig 1. In this figure, the first step is to download the dump file, where we

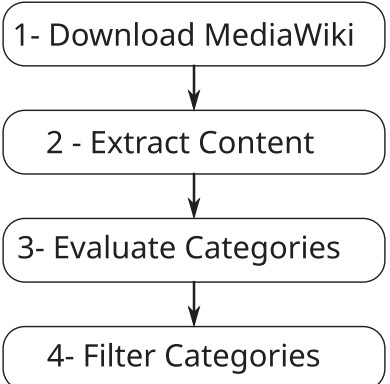

**Fig 1. Steps to prepare the WikiNews dataset.**

**Table 1. WikiNews categories.**

|  | Count |
|---|---|
| Crime, Law and Justice | 1,961 |
| Sport | 3,451 |
| Economy and business | 1,151 |
| Politics | 804 |
| Health | 1,768 |
| Total | **9,135** |

use the May 1, 2022 file. The second step is content extraction, where the marked text is converted to plain text, which was done with BeautifulSoup [33] and MWParserFromHell [34]. The categories, in the MediaWiki syntax, are expressed using a particular markup in the page's content, making it possible to link the same news to several categories. In the extracted data, most news was related to more than one category, such as politics and economics simultaneously. In step 3, we searched for the most frequent and distinct categories, that is, those with few articles belonging simultaneously to more than one of the selected categories. This was done to choose the categories that would be included in the training set.

The last step in preparing the dataset is filtering news from selected categories. News that belonged to more than one of the selected categories has been removed. As a result, a dataset with 9,135 news stories was created, as shown in Table 1, where the categories translated into English are displayed and the news is not evenly distributed around the categories. The code used to extract the Portuguese news, as well as for pre-processing it are accessible [35].

## Experimental setup

This section presents the datasets used in the experiments, the vectorization models and the classifiers used for news categorization.

### Datasets

Three datasets were used in the experiments. The first was WikiNews, which we described in Section News dataset. This data is public and the code used for select Portuguese news and pre-processing it are available in the author GitHub [35]. The second was the Folha dataset. Many of the categories in this database have been eliminated since they have few entries. We also removed articles labeled with categories related to the newspaper section rather than the news content, such as 'columns' and 'illustrated'. The resulting dataset has 96,819 news divided into power (22,022), market (20,970), sport (19,730), the world (17,130), and every day (16,967).

The third dataset was AGNews, which is a collection of more than 1 million news items, containing among other data, titles, descriptions, and the category. The subset we are using was proposed by [36] and can be found in the hugging face library [37]. This dataset consists of 120,000 articles in the training set and 7,600 in the test set divided into 4 balanced categories: world, sports, business, and sci/tech. Since we are interested in Portuguese content, we translate the content using LibreTranslate [38].

Table 2 presents a summary of the datasets. AG News data is divided into two parts, one for training and one for testing. Following this same strategy, we also split the datasets into training and testing parts, using 10% of the total records for the test part. The parts were randomly divided, keeping class ratios, and to enable the replication of results, we provide identification of the articles contained in each portion [39].

**Table 2. Datasets description.**

|  | **WikiNews** | **Folha SP** | **AG News(pt)** |
|---|---|---|---|
| Samples | 9,135 | 96,819 | 127,600 |
| Categories | 5 | 5 | 4 |
| Balanced | No | No | Yes |
| Vocabulary | 55,236 | 235,239 | 104,998 |
| Text length | 191 | 370 | 31 |

We also monitor training and avoid overfitting by using a validation set. This validation set was produced with five-part cross-validation using the Stratified Cross Validation method of the sklearn library [40]. All models used the same input data partitions. In this way, all experiments were repeated five times, each time with a subset of the data, and evaluated with the test set, which did not participate in training at any point.

We've also included some statistics about the vocabulary of the datasets in Table 2. Unlike the preprocessing we used in the classification, we count all words as they appear, but we eliminate any words with characters other than letters or hyphens. We chose this approach to evaluate the number of words with their inflections, different from what will be presented in Section Feature extraction, where methods for removing inflections and composing dictionaries will be included. The 'Vocabulary' row is the number of unique words and the 'Text length' row is the average number of words per news item. The length of the documents varied significantly, as indicated in Fig 2.

## Feature extraction

The procedures applied for data preparation are described in Fig 3. The BERT network requires less preparation, using raw data. For the other methods, we start with preprocessing

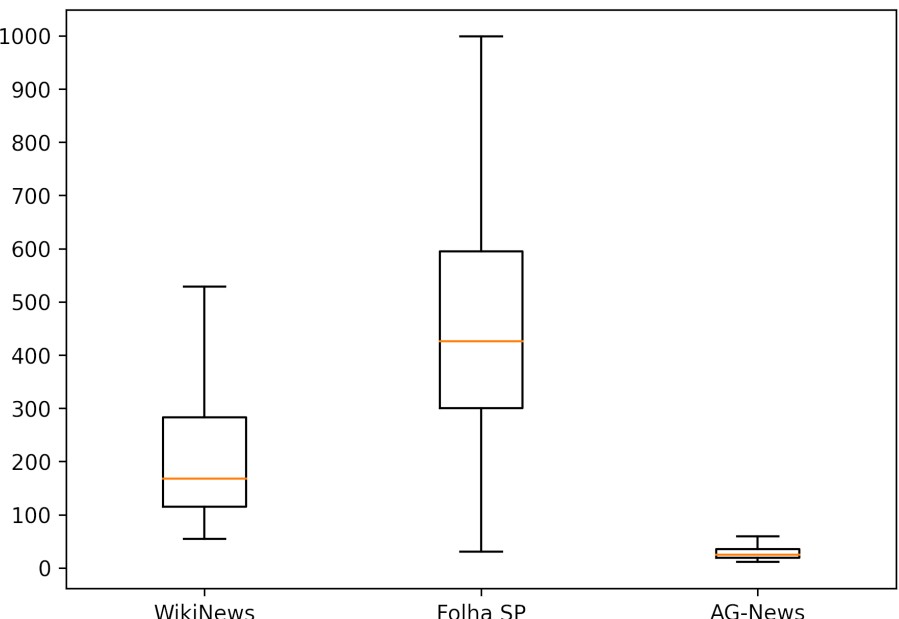

**Fig 2. Number of words per document.**

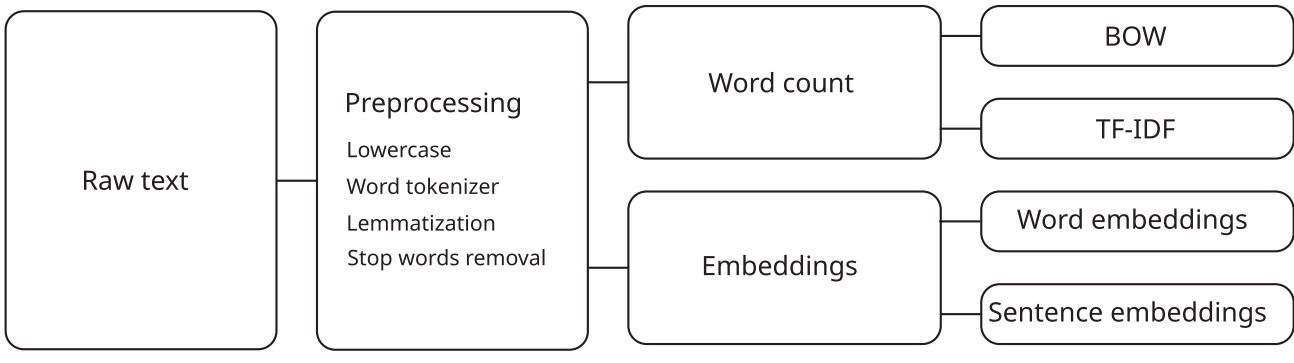

**Fig 3. Dataset preprocessing.**

to simplify the classification by reducing variations among words. For this, the texts are converted to lowercase, the terms are separated with NLTK [41], and the words are converted to their non-inflected form with the spaCy [42] library. Both libraries were configured for Portuguese, and we use the spaCy database optimized for precision (*pt_core_news_lg*). Finally, stop words are also removed with the library NLTK, which also has a database of stop words in Portuguese.

The preprocessing result was used in two ways, which are through word count or word embeddings. When using word count, a dictionary was created composed of the most frequent terms, which was experimentally defined for the size of 2,000 unique words. Using this dictionary, we count the number of occurrences of each term to produce Bag-of-Words (BoW) representations. Also with this dictionary, we generate the TF-IDF representation, which was calculated with the sklearn [43], library.

For embedding representations we use fastText [44], which provides pre-trained models in Portuguese (*cc.pt.300.bin*). We chose this representation due to the possibility of representing subwords, which in addition to allowing morphological variations, also reduces the problem of out-of-vocabulary words. Furthermore, this library has a method for producing sentence embeddings, which is done by averaging the vectors normalized with the $l2$ norm.

## Classifiers

Support Vector Machine (SVM) is a classic machine learning method and one of the most common algorithms found for text classification. We use the implementation of sklearn with a kernel of the Radial Basis Function type. Convolutional Neural Networks (CNN) is a feed-forward deep learning method that has been successfully applied to image and text classification. In this network it is possible to configure an arbitrary number of kernels, which are applied as a sliding window on the input sequence, producing representations in the form of a feature map. The features extracted by the convolutional layers are used by later layers of the network, in our case for classification. We are using a configuration similar to [45], as shown in Fig 4.

In this network, words are represented as word-embeddings and pre-processed values can be used, such as those presented in Section Feature extraction or, following the implementation with PyTorch [46], can be trained using the dataset itself. Convolutional filters are applied over this chain of word embeddings to identify features. In this step, kernels of length between 2 and 5 were combined. Each kernel produces a chain of output activations, where we apply a max-pooling operation, resulting in a single activation value per kernel. We also included a Dropout layer, which in our tests resulted in more consistent results with reduced fluctuation between experiments. Finally, the network output is produced with a softmax layer.

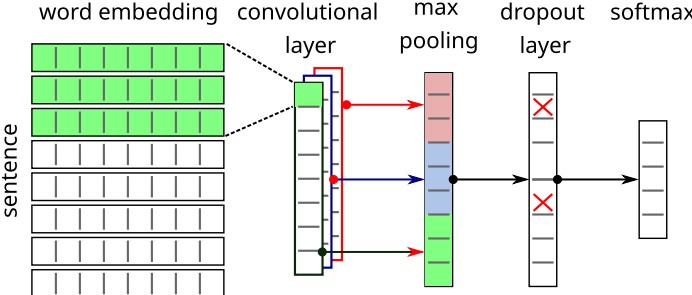

**Fig 4. CNN network.**

The Deep Jointly-Informed Neural Networks (DJINN) [47] aims to combine neural networks' powerful representational capabilities with decision trees' ease of use. Numerous parameters of a neural network, particularly a Multilayer Perceptron network, need to be defined and affect the output results. Among them are the number of layers, the number of neurons per layer, and the weight of connections between neurons. A decision tree, on the other hand, has fewer parameters, although it sometimes performs worse than a neural network.

To simplify the use of neural networks, the DJINN network operates in two stages. A decision tree is used for training in the first stage, and a method is suggested for using the resulting tree structure to define the neural network topology and establish the connections between neurons, which are then fine-tuned. This procedure can be applied both to a single tree or an ensemble of trees, such as those produced by the Random Forest method and in these cases, the result is also an ensemble of neural networks.

The last method we evaluated was the BERT network, which is an encoder-type Transformer. The BERT network is composed of multiple layers, each layer containing a feed-forward network and self-attention heads. This network was trained with huge datasets, which the network used to extract information about the language's semantic features, and the resulting parameter set is distributed as a pre-trained network. Using this pre-trained network, it is possible to add a new output layer and tune the network for a variety of problems, through a process known as fine-tuning.

## Results and discussion

This section presents the experimental setup and the classification results on Portuguese News datasets. We present a trade-off between accuracy and processing time.

### Experiment setup

The CNN network contains several configuration parameters that were determined through experimentation. The parameters we consider are listed in Table 3, totaling 36 possible

**Table 3. Parameters optimized for CNN.**

|  | Values |
| --- | --- |
| Convolutional kernels | 64, 128, 256 |
| Word embeddings | 32, 64, 128, 256 |
| Dropout rate | 0.1, 0.2, 0.3 |

**Table 4. News classification: f1-score.**

| Input | Features | Classifier | WikiNews | Folha SP | AG News(pt) |
|---|---|---|---|---|---|
| Preprocessed | BoW | SVM | 0.804 | 0.933 | 0.897 |
| Preprocessed | TF-IDF | SVM | **0.827** | **0.946** | 0.898 |
| Preprocessed | TF-IDF | DJINN | 0.783 | 0.944 | 0.901 |
| Preprocessed | fastText(sentence) | SVM | 0.812 | 0.931 | 0.890 |
| Preprocessed | fastText(sentence) | DJINN | 0.806 | 0.929 | 0.879 |
| Preprocessed | tokens | CNN | 0.806 | 0.926 | 0.887 |
| Preprocessed | fastText(tokens) | CNN | 0.802 | 0.925 | 0.894 |
| Raw | | BERT | 0.821 | 0.945 | **0.918** |

combinations of values. As mentioned previously we are using four convolutional kernel lengths, and the first parameter of this table is the number of kernels per length. We are also considering the length of the word embeddings and the dropout layer removal rate. We used the WikiNews dataset in all experiments. Only the training set was employed, so the definition of the network topology was not influenced by the test set. Experiments were performed following a five-part cross-validation approach and the best configuration has been applied to all datasets.

We also evaluated the use of pre-trained word embeddings. In this case, the length of the word embeddings was 300, which is the default of the fastText model we use. We allowed the network to calibrate the word embeddings during training and evaluated the convolutional kernels and drop-out rate settings. Thus, we have two network configurations, where the first uses pre-trained embeddings with 64 convolutional kernels and a drop-out rate of 0.1, and the second trains word-embeddings of length 256, with 128 convolutional kernels and a drop-out rate of 0.3.

The calibration parameters for the DJINN network were established experimentally using the WikiNews dataset before being applied to all datasets. In their original article, the authors argue that the use of an ensemble of networks benefits the accuracy of the network. In our evaluation, we found that the accuracy of the network showed little variation after employing 50 networks, which was the number of networks defined. In addition, we defined that each decision tree must have a maximum of 4 levels and we do not use dropout layers.

To evaluate the BERT network, we started with the pre-trained network BERTimbau [48] Base, which was trained with Brazilian Web as Corpus, which is a dataset crawled and filtered from the internet, containing 3,5 million pages and 2,7 billion tokens. Using the Hugging Face implementation, an output SoftMax layer was added and fine-tuned.

We monitor training on the CNN and Bert networks using the validation set produced with cross-validation, stopping when the error was no longer declining. After each training, the models were evaluated with the test set. In this approach, the test set is only used for evaluating the final performance and is never used in training. The results are represented in Table 4, and correspond to the median values of unweighted average f1-score in the problem categories.

## Classification results

The results are presented in Table 4, where the best results are highlighted in bold with a gray background. Using the Kruskal-Wallis H-test to analyze the data from each dataset, we discovered differences that were significant with a 95% confidence level across all datasets. For this reason, using the same confidence level, we conducted a post hoc analysis with Wilcoxon rank-sum statistic to compare methodologies. Using this analysis, we mark with a gray

background those results where we do not find sufficient support to conclude that they differ from the best-performing method, and may even be equivalent.

The proposed dataset, WikiNews, had the lowest f1-score. Probably because in addition to the lower number of records, the proximity of the contents may have influenced the outcome, because even removing news that belonged to more than one of the selected categories, the themes remain close. When analyzing the confusion matrices, as in the example shown in Fig 5, we noticed that the methods had difficulty in distinguishing articles on "Politics" due to the proximity to "Crime, Law and Justice", "Economy, and Business" and "Heath".

To better evaluate this classification challenge in the WikiNews dataset, we chose a few cases for an examination utilizing Local Interpretable Model-Agnostic Explanations [49] (LIME). With this tool, it is possible to analyze the decision of a classifier in a simplified way. For this, in text classification, LIME creates a collection of slightly modified versions of a text to be investigated and uses the classifier to assign labels. In this way, it is possible to determine which changes were most important for changing the categories. The result is a set of terms related to a positive or negative score, indicating whether a term contributes or opposes the assigned classification and the intensity of this influence.

We used the BERT classifier and some samples from the "Politics" category, which had been incorrectly classified. The first selected article was incorrectly classified as health, and was related to political interference in the fight against the Covid-19 epidemic [50]. This article contains terms such as "health", "medicines" and "cases" that were important for the incorrect classification. Another case also classified as health was the article that reported the death of federal deputy Enéas Carneiro [51]. The term "leukemia", which was the cause of death and is more frequently used in health news, was important in this classification. The article [52] has

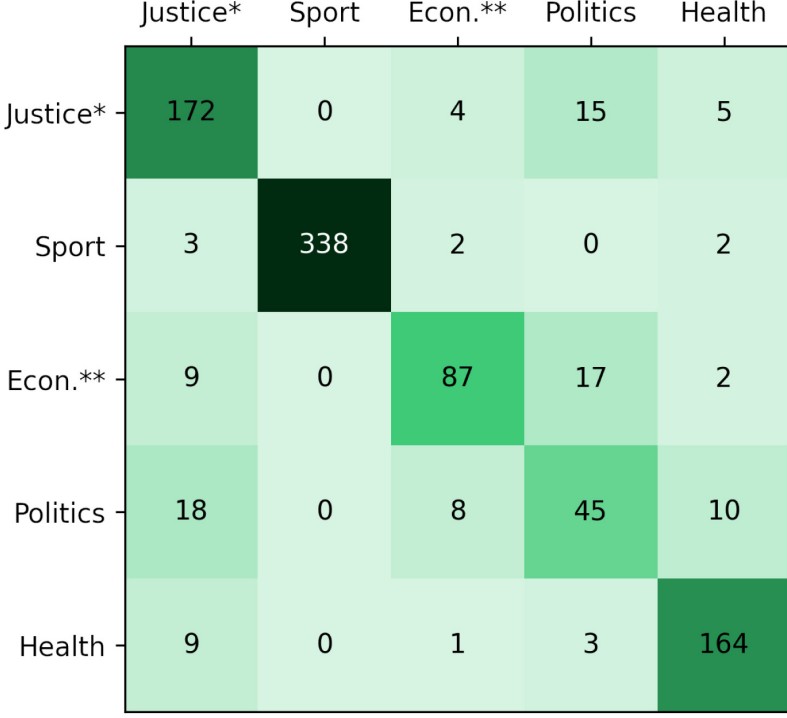

**Fig 5. WikiNews confusion matrix with BERT.** * Crime, Law and Justice, ** Economy and business.

been categorized as "Crime, Law and Justice" and describes that the president of Honduras was detained by the military before the country started a public consultation to reform the constitution. In this article, the most relevant terms for misclassification were "detained" and "prosecution".

We also used LIME to compare the BERT and SVM TF-IDF classifiers, which were the classifiers with the best results. For this, we evaluated articles in which these classifiers disagreed, and we looked for clues about how the errors occurred. The first case we highlight is the article [53] which reports on elections in Zimbabwe, and which belongs to the 'Politics' category but was incorrectly classified by the SVM as 'Health'. One of the most important terms for this classification is the word "die", which is present in "Vote for Mugabe, or die of starvation". Evaluating the results with LIME, the BERT network was little influenced by this term, perhaps because of its inflection. In SVM, preprocessing removed the inflection of the term in Portuguese, bringing the spelling closer to the same found in health articles. In its original form, as applied in the BERT network, this word is a verb in the imperative mood, which is when a reaction is required from the interlocutor, perhaps less common in health articles.

Another case is the article [54], which belongs to the "Crime, Law and Justice" category and was incorrectly classified by the BERT network as "Politics". The article reports the removal of a deputy from the position of president of his party and is correctly classified as "crime, law and justice" since the motivation for the removal was the fulfillment of a court order [55]. However, in the article, the reason for the removal is not mentioned, and it is not possible to infer without external data that this is a case of justice. Compared with the SVM classifier, the most important terms to classify as justice were "deputy" and "party", which are not terms directly linked with justice, but must be frequent in the news of this type in the data set.

We also included, in Table 5, the processing times, which were measured using an Intel Xeon 3.2 GHz processor with 32 GB of RAM. The training times of the BERT and DJINN

**Table 5. News classification: Processing time.** The training times are at the top of the table, and the test time is at the bottom.

| Input | Features | Classifier | WikiNews | Folha SP | AG News(pt) |
|---|---|---|---|---|---|
| Preprocessed | BoW | SVM | 0:00:30 | 0:52:55 | 2:57:06 |
| Preprocessed | TF-IDF | SVM | 0:00:13 | 0:43:26 | 0:28:43 |
| Preprocessed | TF-IDF | DJINN | 0:30:58* 10:50:00** | 7:15:00* 131:00:00** | 6:20:00* 213:00:00** |
| Preprocessed | fastText(sentence) | SVM | 0:00:02 | 0:03:48 | 0:11:16 |
| Preprocessed | fastText(sentence) | DJINN | 0:09:59* 0:37:00** | 0:40:00* 03:50:00** | 0:55:00* 05:50:00** |
| Preprocessed | tokens | CNN | 0:04:26 | 0:49:48 | 1:04:46 |
| Preprocessed | fastText(tokens) | CNN | 0:03:06 | 0:33:32 | 0:38:51 |
| Raw | | BERT | 0:11:00* 6:49:00** | 1:27:00* 53:50:00** | 2:16:00* 77:15:00** |
| Preprocessed | BoW | SVM | 0:00:06 | 0:06:52 | 0:09:30 |
| Preprocessed | TF-IDF | SVM | 0:00:02 | 0:03:00 | 0:00:40 |
| Preprocessed | TF-IDF | DJINN | 0:00:08 | 0:01:40 | 0:01:17 |
| Preprocessed | fastText(sentence) | SVM | 0:00:01 | 0:00:50 | 0:01:14 |
| Preprocessed | fastText(sentence) | DJINN | 0:00:09 | 0:00:04 | 0:00:03 |
| Preprocessed | tokens | CNN | 0:00:01 | 0:00:11 | 0:00:13 |
| Preprocessed | fastText(tokens) | CNN | 0:00:01 | 0:00:10 | 0:00:07 |
| Raw | | BERT | 0:02:58 | 0:33:12 | 0:10:13 |

* Time on GPU

** Estimated time on CPU

networks were measured during GPU Tesla T4 training in Google Colab. We added the estimated training time using the same system used in the other methods for comparison. The training times are at the top of the table, and the application times for predicting the values of the test sets are at the bottom. To simplify the comparison, we present the times measured in all training iterations, despite the fact that in most cases a result of an intermediate training step was used, depending on the result observed with the validation set. We can notice in the table that DJINN + TF-IDF is the costly approach, followed by BERT.

## Discussion

Next, we discuss the pros and cons of the mentioned algorithms based on their accuracy and running time.

- SVM with BoW:
  Pros: Provides reasonable accuracy across all data sources.
  Cons: Despite having decent accuracy, it can be outperformed by other more advanced algorithms.

- SVM with TF-IDF:
  Pros: Has generally high accuracy, especially for WikiNews and Folha SP data sources.
  Cons: Has lower accuracy compared to BERT for the AG News(pt) data source.

- DJINN with TF-IDF:
  Pros: It presents a competitive accuracy, especially for the Folha SP and AG News(pt) data sources.
  Cons: Takes a long time to train, especially for Folha SP and AG News(pt) data sources. Furthermore, its accuracy is surpassed by BERT.

- SVM with fastText (sentence):
  Pros: Provides reasonable accuracy for all data sources.
  Cons: Does not outperform other more advanced algorithms like BERT and DJINN.

- DJINN with fastText (sentence):
  Pros: Has acceptable accuracy for all data sources.
  Cons: Requires considerable training time, especially for Folha SP and AG News(pt) data sources. Accuracy is also lower than BERT.

- CNN with tokens:
  Pros: Provides decent accuracy for all data sources.
  Cons: Does not achieve the same accuracy as BERT and DJINN.

- CNN with fastText (tokens):
  Pros: Has acceptable accuracy for all data sources.
  Cons: Does not match the accuracy of BERT and DJINN.

- BERT (raw):
  Pros: Achieves high accuracy for all data sources.
  Cons: Takes longer to train and infer, especially for Folha SP and AG News(pt) data sources.

In summary, the BERT algorithm has high accuracy across all data sources but is slower in terms of running time. For a more balanced approach between accuracy and execution time, the algorithms SVM with TF-IDF can be considered, as they present reasonable accuracy and relatively fast execution time. However, the final choice will depend on the project's specific needs and constraints.

## Conclusions

In this work, we presented a comparison of news categorization techniques, where different techniques for extracting attributes and building models were considered. In our experiments, conventional feature extraction and classification techniques, especially the combination of TF-IDF with SVM, showed reasonable results. Considering that these methods have the lowest computational cost, they can be an option in scenarios that involve large volumes of data or that need results in real-time.

MLP network training is computationally intensive, especially with a high number of attributes. By training multiple networks of this type, the DJINN network presented the longest training time. The effect of the number of attributes can be observed by comparing the difference in processing times when fastText and TF-IDF are applied in the attribute extraction step. With fastText, each sentence results in a set of only 300 attributes and it was much faster than TF-IDF where we used a dictionary with 2,000 words.

BERT was the approach that consistently produced the best results, with results statistically indistinguishable from the top methods in WikiNews, and FolhaSP and achieving the best result in AGNews. In addition, it is the simplest method to be applied because it eliminates the preprocessing step. The availability of a pre-trained model in Portuguese, BERTimbau, also greatly simplified the experiments. Considering the computational cost, this network is preferable in scenarios where accuracy is a critical factor.

## Author Contributions

**Conceptualization:** Klaifer Garcia, Lilian Berton.

**Data curation:** Klaifer Garcia.

**Funding acquisition:** Pedro Shiguihara.

**Investigation:** Pedro Shiguihara.

**Methodology:** Klaifer Garcia, Lilian Berton.

**Software:** Klaifer Garcia.

**Supervision:** Lilian Berton.

**Validation:** Klaifer Garcia.

**Writing – original draft:** Klaifer Garcia.

**Writing – review & editing:** Pedro Shiguihara, Lilian Berton.

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
