## [Decision Letter · Decision Letter 0]

4 May 2023

PONE-D-22-26478Portuguese news classificationPLOS ONE

Dear Dr. Berton,

Thank you for submitting your manuscript to PLOS ONE. After careful consideration, we feel that it has merit but does not fully meet PLOS ONE’s publication criteria as it currently stands. Therefore, we invite you to submit a revised version of the manuscript that addresses the points raised during the review process.

We look forward to receiving your revised manuscript.

Kind regards,

Diego Raphael Amancio

Academic Editor

PLOS ONE

Journal Requirements:

2. In your Methods section, please include additional information about your dataset and ensure that you have included a statement specifying whether the collection method and the public sharing of the collected dataset complied with the terms and conditions for the website.

Reviewers' comments:

Reviewer's Responses to Questions

**Comments to the Author**

1. Is the manuscript technically sound, and do the data support the conclusions?

Reviewer #1: Partly

Reviewer #2: Partly

2. Has the statistical analysis been performed appropriately and rigorously? 

Reviewer #1: No

Reviewer #2: N/A

3. Have the authors made all data underlying the findings in their manuscript fully available?

Reviewer #1: Yes

Reviewer #2: Yes

4. Is the manuscript presented in an intelligible fashion and written in standard English?

Reviewer #1: No

Reviewer #2: Yes

5. Review Comments to the Author

Reviewer #1: Authors presented the research on Portuguese news classification. After reading the whole manuscript, following observations have been made.

I. The structure and layout of the paper should be consistent throughout the manuscript.

2. the usage of the English Language should be improved in the whole manuscript.

3. The technical depth of the manuscript is weak and should be enhanced.

4. Result section should be improved.

5. Reference section needs to be strengthened with the recent one. Following references should be incorporated in the manuscript.

[1] "Fake News Classification Based on Content Level Features" Applied Sciences 12, no. 3: 1116. https://doi.org/10.3390/app12031116

[2]Digital Transparency and Open Data. In: Farazmand A. (eds) Global Encyclopedia of Public Administration, Public Policy, and Governance. Springer, Cham, Springer Nature Switzerland AG 2020.

Reviewer #2: The authors present the process of building a Portuguese news dataset, to encourage research in news categorization in that language. The main motivation given by the authors is that there is a lack of Portuguese news datasets, slowing down the task's progress. While other resourceful languages are advancing in the task, the number of works produced in Portuguese is significantly lower, and the categorization of news in Portuguese lacks a comparison of approaches that incorporates new state-of-the-art technologies. In this context, the other contribution of the paper is presented: “a comparison of different text representation (BoW, TF-IDF, Embedding) and classification (SVM, DJINN, CNN, BERT) methods for Portuguese news, exploring parameterization and processing time;”

The final experimental evaluation presents the results of the proposed dataset (WikiNews) along with Folha de SP and the AG News (translated), showing that BERT achieved the best results, while CNN did not do very well. In terms of cost-effectiveness, the combination of TF-IDF with SVM can be an alternative in scenarios where there is a need for reasonable results with low computational limits or that need results in real-time.

I believe the authors have done good work. The article is well-written, easy to understand, and addresses an interesting task. The motivation is genuine and well-founded. The contributions are valid and important for several tasks, not only for news categorization. Concerning the benchmark, it is very important, because the efforts of the researchers will be able to take place on a strong foundation.

My main concern is with how the datasets have been trained. The authors state that they use the test sets as early stopping criteria during CNN and BERT training. And the validation sets were used to calculate the results of the experiments. But, from what I understood, in the case of AG News, only the test set was used for both. If it was used in this way, that is a serious problem. During training, the model should not have any influence from the test set, even indirectly. The way it is being used is influencing the choice of model parameters (since training stops when the parameters are optimized for the test set) even if it does not directly access the objects in the test set. My suggestion to the authors is to correct this flaw, for example, using part of the training set as validation. With this correction, the results will be much more reliable, even if there are no considerable changes in the results.

I have some minor suggestions/questions, to improve the text:

Line 108: “The category information was separated from the text to be analyzed in step 3, where we chose the most common categories with the smallest category intersection or the smallest number of pages that belong to multiple categories at the same time”. – That sentence is not clear. I don't understand it.

Line 153: ScyPy or Spacy?

In table 3: I suggest highlighting the chosen parameter values;

Line 278: “A factor that may have influenced this difference was the size of the embeddings, which was larger than the pre-trained ones.” – Were the pre-trained embeddings kept static during training? This may also justify the slightly worse result.

What percentages are used for training, testing, and validation of each dataset?

Perhaps using a heatmap could improve the visualization of Table 6.

Once again, I believe the authors have done good work and this is a nice and standard experiment design. An additional suggestion to the authors is to try to enrich the discussion by bringing in the most relevant features (words) for each category. Is there a word or set of words that are most relevant for the category classification? I suggest authors read the following paper:

Marco Ribeiro, Sameer Singh, and Carlos Guestrin. 2016. “Why Should I Trust You?”: Explaining the Predictions of Any Classifier. In Proceedings of the 2016 Conference of the North American Chapter of the Association for Computational Linguistics: Demonstrations, pages 97–101, San Diego, California. Association for Computational Linguistics.

6. PLOS authors have the option to publish the peer review history of their article (what does this mean?). If published, this will include your full peer review and any attached files.

Reviewer #1: No

Reviewer #2: No

---

## [Author Response · Author response to Decision Letter 0]

27 Jun 2023

Original Manuscript ID: PONE-D-22-26478

Original Article Title: “Portuguese news classification ”

To: IEEE Access Editor

Re: Response to reviewers

Dear Editor,

Thank you for allowing a resubmission of our manuscript, with an opportunity to address the reviewers’ comments.

We are uploading (a) our point-by-point response to the comments (below) (response to reviewers), (b) an updated manuscript with blue and red highlighting indicating changes (Supplementary Material for Review), and (c) a clean updated manuscript without highlights (Main Manuscript).

Best regards,

Klaifer Garcia and Lilian Berton

Response to Reviewers: 

Thank you for the timely review and very constructive comments! We have addressed all the suggestions including more content in the paper and believe it is much improved. We also have put together a point-by-point set of responses below. 

Reviewer's Responses to Questions

1. Is the manuscript technically sound, and do the data support the conclusions?

Reviewer #1: Partly

Reviewer #2: Partly

Author response: Thank you again for your constructive remarks. Constructive criticism plays a key role in advancing scientific research, and we hope the clarifications and updates help provide a fuller perspective on the manuscript. Regarding the technical soundness of the manuscript, we would like to highlight that every effort was made to ensure the quality and accuracy of the research. The study was conducted following rigorous scientific standards and established methodologies.

2. Has the statistical analysis been performed appropriately and rigorously?

Reviewer #1: No

Reviewer #2: N/A

Author response: We changed the experiments and ran them again with cross validation. This approach allowed us to obtain results suitable for conducting statistical tests. Specifically, we performed the "Kruskal-Wallis one-way analysis of variance" to analyze the differences among multiple groups, and subsequently conducted post hoc comparisons using the "Mann-Whitney U test." These tests helped us assess the significance of the observed data and draw conclusions based on the statistical analysis.

3. Have the authors made all data underlying the findings in their manuscript fully available?

Reviewer #1: Yes

Reviewer #2: Yes

Author response: Yes, all the data is fully available. The code and instructions to replicate the dataset is free available in the first author github: https://github.com/Klaifer/PortugueseNewsDataset

4. Is the manuscript presented in an intelligible fashion and written in standard English?

Reviewer #1: No

Reviewer #2: Yes

Author response: We've revised the entire article, re-editing some parts for clarity. Careful attention has been given to ensure that the language used is clear, coherent, and conforms to the conventions of standard English. The text is organized logically, with a clear introduction, well-structured paragraphs, and a coherent conclusion. Additionally, the manuscript has undergone proofreading and editing to correct any grammatical or typographical errors, further enhancing its readability. 

Reviewer #1: Authors presented the research on Portuguese news classification. After reading the whole manuscript, following observations have been made.

Concern #1: The structure and layout of the paper should be consistent throughout the manuscript.

Author response: We updated the manuscript by rewriting some sections to make clear our contributions and to keep the same layout.

Concern #2: The usage of the English Language should be improved in the whole manuscript.

Author response: Thank you for the comment, we revised the manuscript and the English writing with respect to grammar and sentence structure, clarity and coherence. 

Concern #3: The technical depth of the manuscript is weak and should be enhanced.

Author response: We revised the methodology and experimental design to ensure they are detailed and clearly explained. We also review the data analysis and interpretation to make them thorough and robust. Our results and discussion are also adequately presenting the findings. We improved the experiments with the inclusion of the statistical tests. The CNN network was also improved, using a structure similar to one of the references [A Sensitivity Analysis of (and Practitioners' Guide to) Convolutional Neural Networks for Sentence Classification].

Concern #4: Result section should be improved.

Author response: We presented the results in a logical order, with clarity and precision. We use tables and figures to enhance the presentation. We included statistical tests to support the findings. We also discuss the implications of the findings and provide connections to the research objectives or existing literature. 

Concern #5: Reference section needs to be strengthened with the recent one. Following references should be incorporated in the manuscript.

[1] "Fake News Classification Based on Content Level Features" Applied Sciences 12, no. 3: 1116. https://doi.org/10.3390/app12031116

[2] Digital Transparency and Open Data. In: Farazmand A. (eds) Global Encyclopedia of Public Administration, Public Policy, and Governance. Springer, Cham, Springer Nature Switzerland AG 2020.

Author response: Thank you for the suggestion, we included the reference [1]. The reference [2] unfortunately we could not find it.

Reviewer #2: The authors present the process of building a Portuguese news dataset, to encourage research in news categorization in that language. The main motivation given by the authors is that there is a lack of Portuguese news datasets, slowing down the task's progress. While other resourceful languages are advancing in the task, the number of works produced in Portuguese is significantly lower, and the categorization of news in Portuguese lacks a comparison of approaches that incorporates new state-of-the-art technologies. In this context, the other contribution of the paper is presented: “a comparison of different text representation (BoW, TF-IDF, Embedding) and classification (SVM, DJINN, CNN, BERT) methods for Portuguese news, exploring parameterization and processing time;”

The final experimental evaluation presents the results of the proposed dataset (WikiNews) along with Folha de SP and the AG News (translated), showing that BERT achieved the best results, while CNN did not do very well. In terms of cost-effectiveness, the combination of TF-IDF with SVM can be an alternative in scenarios where there is a need for reasonable results with low computational limits or that need results in real-time.

I believe the authors have done good work. The article is well-written, easy to understand, and addresses an interesting task. The motivation is genuine and well-founded. The contributions are valid and important for several tasks, not only for news categorization. Concerning the benchmark, it is very important, because the efforts of the researchers will be able to take place on a strong foundation.

Concern #1: My main concern is with how the datasets have been trained. The authors state that they use the test sets as early stopping criteria during CNN and BERT training. And the validation sets were used to calculate the results of the experiments. But, from what I understood, in the case of AG News, only the test set was used for both. If it was used in this way, that is a serious problem. During training, the model should not have any influence from the test set, even indirectly. The way it is being used is influencing the choice of model parameters (since training stops when the parameters are optimized for the test set) even if it does not directly access the objects in the test set. My suggestion to the authors is to correct this flaw, for example, using part of the training set as validation. With this correction, the results will be much more reliable, even if there are no considerable changes in the results.

Author response: Thank you very much for pointing out this problem. We run the experiments again using k fold cross validation. With the new configuration, the training monitoring was carried out using the part extracted from the cross-validation and the test set was used only in the final evaluation, without any participation in the training. We also execute statistical tests to evaluate the results.

Concern #2: I have some minor suggestions/questions, to improve the text:

Line 108: “The category information was separated from the text to be analyzed in step 3, where we chose the most common categories with the smallest category intersection or the smallest number of pages that belong to multiple categories at the same time”. – That sentence is not clear. I don't understand it.

Author response: We rewrite it.

Line 153: ScyPy or Spacy?

Author response: We rewrite it.

In table 3: I suggest highlighting the chosen parameter values;

Author response: There are now two CNN network configurations. As a result, the table with the highlighted parameters may no longer be as straightforward to comprehend.

Line 278: “A factor that may have influenced this difference was the size of the embeddings, which was larger than the pre-trained ones.” – Were the pre-trained embeddings kept static during training? This may also justify the slightly worse result.

Author response: Indeed, you were right, thank you. We changed it to allow pre-trained embeddings to be refined, and the results improved. We have included this change in the network description.

Concern #3: What percentages are used for training, testing, and validation of each dataset?

Author response: In this new version of the document, we indicate the separation of 10% of the data for testing. We also change it so that the test partition has the same proportion of samples per class as the full set. The validation set is now produced with 5-part cross-validation, so the set has 20% of the training data.

Concern #4: Perhaps using a heatmap could improve the visualization of Table 6.

Author response: Thank you for the suggestion! We changed the table to heatmap.

Concern #5: Once again, I believe the authors have done good work and this is a nice and standard experiment design. An additional suggestion to the authors is to try to enrich the discussion by bringing in the most relevant features (words) for each category. Is there a word or set of words that are most relevant for the category classification? I suggest authors read the following paper: Marco Ribeiro, Sameer Singh, and Carlos Guestrin. 2016. “Why Should I Trust You?”: Explaining the Predictions of Any Classifier. In Proceedings of the 2016 Conference of the North American Chapter of the Association for Computational Linguistics: Demonstrations, pages 97–101, San Diego, California. Association for Computational Linguistics.

Author response: Thank you for the suggestion, we included the analysis suggested by this paper in our work.

---

## [Decision Letter · Decision Letter 1]

27 Dec 2023

Breaking News: Unveiling a New Dataset for Portuguese News Classification and Comparative Analysis of Approaches

PONE-D-22-26478R1

Dear Dr. Berton,

We’re pleased to inform you that your manuscript has been judged scientifically suitable for publication and will be formally accepted for publication once it meets all outstanding technical requirements.

Kind regards,

Diego R. Amancio

Academic Editor

PLOS ONE

Additional Editor Comments (optional):

Reviewers' comments:

Reviewer's Responses to Questions

**Comments to the Author**

1. If the authors have adequately addressed your comments raised in a previous round of review and you feel that this manuscript is now acceptable for publication, you may indicate that here to bypass the “Comments to the Author” section, enter your conflict of interest statement in the “Confidential to Editor” section, and submit your "Accept" recommendation.

Reviewer #1: (No Response)

Reviewer #2: All comments have been addressed

2. Is the manuscript technically sound, and do the data support the conclusions?

Reviewer #1: Partly

Reviewer #2: Yes

3. Has the statistical analysis been performed appropriately and rigorously? 

Reviewer #1: No

Reviewer #2: Yes

4. Have the authors made all data underlying the findings in their manuscript fully available?

Reviewer #1: No

Reviewer #2: Yes

5. Is the manuscript presented in an intelligible fashion and written in standard English?

Reviewer #1: Yes

Reviewer #2: Yes

6. Review Comments to the Author

Reviewer #1: Authors have improved the manuscript from the previous version. Following minor changes are suggested in the manuscript. Author were unable to find the following reference

[2]Digital Transparency and Open Data. In: Farazmand A. (eds) Global Encyclopedia of Public Administration, Public Policy, and Governance. Springer, Cham, Springer Nature Switzerland AG 2020.

The online link is mentioned as below:

https://link.springer.com/referenceworkentry/10.1007/978-3-319-31816-5_3957-1

This reference should be added in the manuscript.

Reviewer #2: The paper introduces a novel news dataset named WikiNews and conducts a series of experiments employing various classifiers and diverse representations to validate and compare the new dataset. The study is interesting, presenting good arguments that contribute to its relevance within the community. I agree with the authors on the importance of having different news datasets with distinct sources and writing styles to mitigate biases.

The experiments are well-designed and align coherently with the paper's objectives. The authors provide essential details regarding the primary parameters and configurations necessary for replication. I particularly appreciate the comprehensive analysis of errors, elucidating possible motivations behind the classifiers' decisions.

Lastly, the inclusion of execution times for the classifiers adds valuable temporal considerations for readers, helping them in selecting the most suitable classifier. Congratulations on your work, and I trust it will be valuable to many researchers. Well done, and I hope it becomes a valuable resource in the field.

7. PLOS authors have the option to publish the peer review history of their article (what does this mean?). If published, this will include your full peer review and any attached files.

Reviewer #1: No

Reviewer #2: No

---

## [Editor Report · Acceptance letter]

17 Jan 2024

PONE-D-22-26478R1 

PLOS ONE

Dear Dr. Berton, 

I'm pleased to inform you that your manuscript has been deemed suitable for publication in PLOS ONE. Congratulations! Your manuscript is now being handed over to our production team.

Kind regards, 

on behalf of

Dr. Diego R. Amancio 

Academic Editor

PLOS ONE